# CHP Engine Anomaly Detection Based on Parallel CNN-LSTM with Residual Blocks and Attention

**DOI:** 10.3390/s23218746

**Published:** 2023-10-26

**Authors:** Won Hee Chung, Yeong Hyeon Gu, Seong Joon Yoo

**Affiliations:** 1Artificial Intelligence Department, Sejong University, Seoul 05006, Republic of Korea; whchung@sejong.ac.kr; 2Computer Science and Engineering Department, Sejong University, Seoul 05006, Republic of Korea; sjyoo@sejong.ac.kr

**Keywords:** engine anomaly detection, convolutional neural network, long short-term memory, residual block, attention mechanism, Bayesian optimization

## Abstract

The extreme operating environment of the combined heat and power (CHP) engine is likely to cause anomalies and defects, which can lead to engine failure; thus, detecting engine anomalies is essential. In this study, we propose a parallel convolutional neural network–long short-term memory (CNN-LSTM) residual blocks attention (PCLRA) anomaly detection model with engine sensor data. To our knowledge, this is the first time that parallel CNN-LSTM-based networks have been used in the field of CHP engine anomaly detection. In PCLRA, spatiotemporal features are extracted via CNN-LSTM in parallel and the information loss is compensated using the residual blocks and attention mechanism. The performance of PCLRA is compared with various hybrid models for 15 cases. First, the performances of serial and parallel models are compared. In addition, we evaluated the contributions of the residual blocks and attention mechanism to the performance of the CNN–LSTM hybrid model. The results indicate that PCLRA achieves the best performance, with a macro f1 score (mean ± standard deviation) of 0.951 ± 0.033, an anomaly f1 score of 0.903 ± 0.064, and an accuracy of 0.999 ± 0.002. We expect that the energy efficiency and safety of CHP engines can be improved by applying the PCLRA anomaly detection model.

## 1. Introduction

As global warming accelerates and emerges as a major problem, many countries are striving to address carbon emissions by implementing various policies, including energy-related regulations, incentives, and research and development subsidies [1]. In South Korea, efforts are being made to transition from traditional thermal power generation to ecofriendly renewable energy; however, this transition is hindered by physical space restrictions, owing to difficulties in securing land, policies focused on quantitative supply, and the economic support required for the transition to eco-friendly power generation technology [2]. Therefore, combined heat and power (CHP) plants have attracted attention as an alternative to conventional thermal power generation for simultaneously achieving the goals of environmental protection and energy efficiency [3]. CHP plants use liquefied natural gas as a fuel to produce and provide heat and electric power simultaneously [4]. CHP plants emit smaller quantities of greenhouse gases and are more efficient than traditional thermal power plants because the heat generated while generating electricity can be used in the absorption system of refrigeration and heating. Despite their reliance on fossil fuels, CHP plants are recognized for their applicability because the fuel used in CHP engines can be replaced with renewable fuel; thus, existing thermal power plants can be converted for eco-friendly application. Therefore, compared with conventional thermal power generation, the power generation efficiency of CHP plants is higher and the environmental impact is smaller [5].

However, CHP engines are operated in high-temperature and high-pressure environments, which increases the risk of mechanical and system anomalies and faults, owing to the strain on components. Poor management of engine anomalies increases fuel consumption with reduced operational efficiency, which results in higher greenhouse gas emissions and increases the risk of sudden engine failure and disruption [6]. Therefore, anomaly management is essential in CHP operation management. Rule-based methods are typically used for anomaly detection in power generator engines; however, these methods are disadvantageous because they limit the detection performance and incur significant costs in the design and development of the rules [7]. Data-based machine learning and deep learning anomaly detection techniques can be utilized to mitigate these disadvantages [8]. In particular, deep learning-based anomaly detection can enhance the safety and efficiency of engine operation by learning abnormal occurrence patterns from the sensor data of operating engines and applying them to actual operations to detect anomalies in advance and provide alarms to power plants.

In this study, we propose an anomaly detection model for CHP engines that has a parallel convolutional neural network (CNN)–long short-term memory (LSTM) residual attention (PCLRA) model, which is a hybrid model of various deep learning algorithms.

### 1.1. Related Works

The detection of anomalies in a power generator engine and turbine can be achieved with classification models using multivariate time series data. The models used for this can be categorized into shallow machine learning, deep learning, and hybrid deep learning.

Shallow machine learning involves finding and learning patterns in large amounts of data. Wang et al. [9] proposed an anomaly detection model for an integrated energy system that included electricity, gas, and heat using a support vector machine (SVM) [10], and they demonstrated the superior performance of the model to statistical models. Lee et al. [11] proposed an anomaly detection model for aircraft using an SVM. Wang et al. [12] demonstrated the performance of a naïve Bayesian-based [13] anomaly detection model for power plant fan systems by comparing the model with random forest (RF) [14] and k-nearest neighbor (KNN) models [15].

While shallow machine learning models train data patterns to derive detection results, deep learning has been used to find and learn important features that affect anomalies in a vast amount of data, and the excellent performance of this method has been demonstrated in numerous studies. In particular, studies have been conducted on the vanishing gradient problem of neural network structures and deep learning has been used to solve relevant problems in various fields [16]. First, there have been anomaly detection studies based on the artificial neural network (ANN) and multilayer perceptron (MLP). Alblawi et al. [17] proposed an anomaly detection model using gas turbine sensor data and ANN [18] and they compared its performance with that of thermodynamic computational models. Amirkhani et al. [19] proposed an anomaly detection model using gas turbine sensor data and ANN and compared its performance with that of MLP and statistical models. Zhou et al. [20] preprocessed sensor data with a spatial transformer network to propose an anomaly detection model for gas turbines using MLP and compared it to a model without a transformer. Additionally, LSTM has been used to learn time series data and perform anomaly detection. Wang et al. [21] used principal component analysis [22] to extract key features and proposed an anomaly detection model using LSTM for aircraft acceleration engines. Liu et al. [23] proposed a Bayesian LSTM [24] model for detecting steam turbine anomalies in nuclear power plants and compared its performance with that of the recurrent neural network (RNN). Li et al. [25] proposed an LSTM model for distributed control system anomaly detection and compared it with ANN and extreme learning machine. RNN-based networks have been widely used for anomaly detection with multivariate time series data. However, because it is important to comprehensively understand multiple sensors and learn spatial features for engine anomalies, CNN-based models have also attracted research attention. Li et al. [26] proposed an anomaly detection CNN model for substations and demonstrated its superior performance to ANN, KNN, and RF models. Shahid et al. [27] used a CNN model to detect engine anomalies and compared it with SVM-, KNN-, and CNN-based models. Lee et al. [28] transformed time-series data into two-dimensional (2D) images, proposed an anomaly detection model for nuclear power plants using two-channel CNN [29], and compared its performance with that of one-channel CNN, the gated recurrent unit (GRU) [30], ANN, and SVM. Zhou et al. [31] proposed an anomaly detection model for micro gas turbines using CNN optimized with extreme gradient boosting and compared its performance with that of MLP [18], the deep belief network, and CNN. Yao et al. [32] proposed an anomaly detection model for a nuclear power plant using simulation data and a model that combines residual blocks [33] and CNN, and they compared it with that of CNN.

Studies have been conducted to improve anomaly detection performance using deep learning and data preprocessing techniques. In particular, hybrid deep learning models are attracting attention because they combine different neural network models to overcome the shortcomings of single models and improve the overall performance. Since CNN and LSTM are effective for extracting spatial and temporal features, respectively, from the data, models that combine CNN and LSTM are superior for training the important features of multivariate time series data. Kong et al. [34] evaluated the performance of a CNN-GRU-based wind turbine anomaly detection model. In addition, studies were conducted to improve the performance by applying an attention mechanism [35] to the CNN-RNN network. Xiang et al. [36] proposed a wind turbine anomaly detection model using CNN-LSTM-AM, in which the CNN, LSTM, and attention mechanism were combined. The model exhibited superior performance to the LSTM, BiLSTM, and CNN-LSTM models. Subsequently, an improved CNN-BiGRU-AM model was proposed and compared with the GRU, CNN-GRU, and CNN-BiGRU models [37].

Models that combine CNN and LSTM are effective for extracting and training spatiotemporal features in multivariate data. However, CNN-LSTM-based anomaly detection models for engines and turbines are mostly serially combined. In serial CNN-LSTM models, the output of CNN is used as the input of LSTM; therefore, temporal features cannot be extracted from the original input data using LSTM. In addition, the loss of spatial information extracted by CNN may occur. To address these limitations, parallel CNN-LSTM (PCL), which combines CNN and LSTM in parallel to directly extract spatiotemporal features from the original input data, and parallel CNN-LSTM attention (PCLA), which combines the PCL and attention mechanism, were proposed [38].

### 1.2. Contribution

In previous studies, hybrid deep learning models exhibited a better anomaly detection performance than shallow machine learning and deep learning models. In this study, we propose a parallel CNN-LSTM residual blocks attention (PCLRA) model that combines the attention mechanism and residual blocks using engine sensor data. We applied the model to the anomaly detection of CHP engines to evaluate its performance. The engine sensor log data is the same as the multivariate time series data, and CNN is used to train the spatial features by analyzing multiple sensor data that occurred at the same time, whereas LSTM is used for training temporal features. In addition, residual blocks and an attention mechanism are applied to compensate for information loss due to the vanishing gradient problem in the CNN-LSTM network.

To our knowledge, this is the first time that parallel CNN-LSTM-based networks have been introduced into the field of CHP engine anomaly detection. In PCLRA, the input data are entered into CNN and LSTM in parallel to extract and train spatial and temporal features. This model allows the spatiotemporal features of the original input data to be trained more effectively compared with models that combine CNN and LSTM in series. Residual blocks are used during this process to compensate for the information loss caused by the vanishing gradient problem in the CNN and LSTM. Additionally, the features extracted from the CNN and LSTM are combined and input into the attention mechanism to focus on important spatiotemporal features. Lastly, the anomaly detection results for the CHP engines are derived using the softmax function. The contributions of this study are as follows:PCLRA is proposed as a model that combines CNN and LSTM in parallel and integrates the residual blocks and attention mechanism. This model is applied to anomaly detection in CHP engines.The performances of the parallel CNN-LSTM models are compared with that of the serial CNN-LSTM models, and Bayesian optimization (BO) [39] is applied to identify the hyperparameter values that optimized the performance of each model.The performances of the hybrid models with residual blocks are compared with that of the hybrid models with the attention mechanism, and BO is applied to identify hyperparameter values that optimized the performance of each model.

The remainder of this paper is organized as follows. Section 2 describes the methodologies used in this study, including the proposed model. Section 3 presents details regarding the experiment, including the data, training, testing procedures, and performance comparison results. Section 4 presents the discussion and Section 5 concludes the study.

## 2. Methods

### 2.1. Overall Framework of Multivariate Time Series Anomaly Detection

The multivariate time series anomaly detection framework presented in this study is shown in Figure 1. The engine sensor multivariate time series data set is preprocessed into train, validation, and test sets for input into the model via the normalization and dataset split. A total of 10 models, including the proposed model PCRLA, are trained to compare the model performance of various network structures, and Bayesian optimization is applied to find the optimal hyperparameters. The models are trained with the train set and evaluated with the validation set to find the best hyperparameter combination. The final selected models are retrained and anomaly detection scores for the test set are derived. The optimal network model is selected via anomaly detection score comparison of a total of 10 models. In this chapter, the basic network elements constituting the proposed model and the PCRLA structure are explained. In addition, nine baseline models and hyperparameter optimization are described.

### 2.2. CNN

In the CHP engine system, multiple sensors are mounted on each part of the engine for management. Each part of the engine is connected with the others, so the collected data are multivariate time series data based on the time and sensors. For multivariate time series forecasting, it is important to identify the non-linear and non-periodic characteristics of the data from short-term and long-term dynamic flows. Although RNN-based models have received attention for time series forecasting, they have limitations in that they only extract the temporal features of multivariate time series data. In one-dimensional CNN (1D-CNN), the kernel moves in the time direction and extracts spatial features, so it is well suited for multivariate time series data. The simple architecture of 1D-CNN is shown in Figure 2 and the process is represented as Equation (1). $x_{i}$ is the input, k is the kernel weight, b is the bias, and f(·) is the activation function. Additionally, $x_{j}$ is the output of the *j*th kernel in the lth convolutional layer.
(1)$$x_{j}^{l}=f\left(\sum_{i\in M_{j}}x_{i}^{l-1}\times k_{ij}^{l}+b_{j}^{l}\right)$$

### 2.3. LSTM

RNN-based models, especially LSTM, play an essential role in sequential data analysis such as natural language and time series data. LSTM is especially designed for capturing long-term temporal dependencies and overcoming vanishing gradient problems. LSTM controls the flow of information using four components: cell, input gate, forget gate, and output gate. The simple architecture of LSTM is shown in Figure 3 and the process is represented as Equations (2)–(7), where $x_{t}$ is the input, $h_{t}$ is the output of the hidden layer, C is the memory cell, W is the weight, and b is the bias. The input gate plays a role in determining how much to add to the current cell state value in order to remember the current information. The value obtained by multiplying the current time value $x_{t}$ by the weight $W_{xi}$ and the value obtained by multiplying the previous time hidden state $h_{t-1}$ by the weight $W_{hi}$ leading to the input gate are added. The input gate result is the value applied as the sigmoid function to the added value in Equation (2). The sigmoid function results in a value between 0 and 1, which is the amount of information that has gone through the process. Then, the tanh function is applied by adding the product of $x_{t}$ and the weight $W_{xg}$, leading to the input gate and the product of $h_{t-1}$ and $W_{hg}$, as the amount of information to remember at time t is derived with it and $g_{t}$ in Equation (3). The forget gate is the process of deciding whether or not to discard past information. The current value $x_{t}$ and the previous hidden state $h_{t-1}$ pass through the sigmoid function in Equation (4). The old cell state is updated using output values of the forget gate and the input gate in Equation (5). Finally, the output gate is the process of determining the output value and plays a role in determining how much of the final cell state value to use. The output gate value is the result of applying $x_{t}$ and $h_{t-1}$ to the sigmoid function and is used to determine $h_{t}$ in Equations (6) and (7).
(2)$$i_{t}=\sigma \left(W_{xi}x_{t}+W_{hi}h_{t-1}+b_{i}\right)$$
(3)$$g_{t}=tanh\left(W_{xg}x_{t}+W_{hg}h_{t-1}+b_{g}\right)$$
(4)$$f_{t}=\sigma \left(W_{xf}x_{t}+W_{hf}h_{t-1}+b_{f}\right)$$
(5)$$C_{t}=f_{t}{}^{\circ}C_{t-1}+i_{t}{}^{\circ}g_{t}$$
(6)$$o_{t}=\sigma \left(W_{xo}x_{t}+W_{ho}h_{t-1}+b_{o}\right)$$
(7)$$h_{t}=o_{t}{}^{\circ}tanh\left(C_{t}\right)$$

### 2.4. Residual Block

CNN exhibits excellent performance in extracting and training important features from multi-dimensional data. Furthermore, the residual block has been proposed to compensate for the vanishing gradient problem in simple-stacked CNN model structures. The residual block allows for forward propagation and backpropagation to be performed immediately in the form of a shortcut connection without going through multiple CNN layers. The residual block is given by Equation (8). The output $y_{n}$ is the result derived by entering the input feature map $x_{n}$ and input update weight $W_{n}$ in the network mapping function F(·). The direct mapping function for the residual connection can be represented by H(·). The final output is the sum of the F(·) and H(·) results. In the proposed model, residual blocks are applied to both CNN and LSTM. The residual block for the CNN consists of convolutional, batch normalization, and the activation function layers, whereas that for the LSTM consists of LSTM, batch normalization, and the activation function layers.
(8)$$y_{n}=F\left(x_{n},W_{n}\right)+H\left(x_{n}\right)$$

### 2.5. Attention Mechanism

Attention mechanism is proposed to solve the vanishing gradient problem that occurs in deep learning models used to train multivariate time-series data. When the input data from the encoder are referred to every time results are detected via the decoder, the attention mechanism is used to focus on the important features instead of referring to all the features at the same importance level. The attention mechanism includes four steps that determine the attention score, attention distribution, attention value, and decoder hidden state. First, we obtain the output at the current time t using the hidden states of the encoder $h_{i}$ and decoder $s_{t}$ by calculating the attention score corresponding to the similarity of all $h_{i}$ and $s_{t}$ using Equation (9). $e^{t}$ is a scalar value consisting of the attention score, as expressed in Equation (10). The attention distribution is obtained by converting this scalar value into a probability distribution by applying the softmax function to $e^{t}$, as given via Equation (11). The attention value is the final output of the attention mechanism and it is the result of multiplying and summing the attention distribution and hidden states, as shown in Equation (12). The weight matrix $W_{c}$ bias $b_{c}$ and tanh function are applied to the attention value to obtain the input ${\tilde{s}}_{t}$ for the last output layer, as given via Equation (13). Finally, in the output layer, the weight vector $W_{y}$ and bias $b_{y}$ are applied to the input ${\tilde{s}}_{t}$, as given via Equation (14), and the softmax function is used to derive the anomaly detection result. In PCLRA, the spatiotemporal features extracted from the CNN and LSTM are input into the attention layer; thus, we focus only on the important features for training.
(9)$$score\left(s_{t},h_{i}\right)=s_{t}^{T}h_{i}$$
(10)$$e^{t}=\left[s_{t}^{T}h_{1},\cdots ,\;s_{t}^{T}h_{N}\right]$$
(11)$$\alpha^{t}=softmax\left(e^{t}\right)$$
(12)$$a_{t}=\sum_{i=1}^{N}\alpha_{i}^{t}h_{i}$$
(13)$${\tilde{s}}_{t}=tanh\left(W_{c}\left[a_{t}\cdot s_{t}\right]+b_{c}\right)$$
(14)$${\hat{y}}_{t}=softmax\left(W_{y}{\tilde{s}}_{t}+b_{y}\right)$$

### 2.6. Parallel CNN-LSTM Residual Blocks Attention

In this study, the PCLRA model is proposed for CHP engine anomaly detection. The system sensor log data of CHP engines is a multivariate time series dataset, which is used as the input data. Identifying the non-linear and non-periodic characteristics of the data from short and long-term dynamic flows is important to multivariate time series forecasting. RNN- and LSTM-based models have been mainly used for time series forecasting, but they are limited to only extracting temporal features of multivariate data. Since CNN can train spatial features of multivariate time series data, models combining CNN with LSTM are excellent for training spatiotemporal features. Accordingly, many serial CNN-LSTM-based models have been proposed. However, for these models, it is difficult to extract temporal features from the original data because the output of the CNN is used as the input of the LSTM and the spatial information extracted from the CNN may be lost. Therefore, we propose PCLRA, which is an advanced structure combining CNN and LSTM in parallel.

In PCLRA, the data are entered into CNN and LSTM in parallel and the spatial and temporal features are extracted. During this process, residual blocks are applied to compensate for the loss of information caused by the vanishing gradient problem. Finally, the attention mechanism is applied to features with different characteristics extracted from the networks and trains the models with a focus on the important features. Thus, the attention value for each component is derived and the normal and anomaly probability values can be determined using the dense layer and softmax function.

Figure 4 shows the detailed structure of the proposed model. The CNN is composed of two layers, each consisting of the activation function and batch normalization. The same padding and size 2 kernel are applied. A residual block is added to extract features directly from the input data without going through the two layers of the CNN. The number of output nodes is set to match the number of last output nodes of the second CNN to derive the final CNN feature values via summation. The LSTM is configured with the same parallel structure as the CNN. The spatial and temporal features derived from the CNN and LSTM, respectively, are concatenated and input into the attention mechanism. Then, the importance of the features is trained, and the normal and anomaly probabilities of the engines are derived as the final output using the softmax function.

### 2.7. Baselines

This section introduces the models used as baselines for the anomaly detection performance comparison with the proposed model: CNN, LSTM, serial CNN-LSTM (SCL), serial CNN-LSTM residual (SCLR), serial CNN-LSTM attention (SCLA), serial CNN-LSTM residual attention (SCLRA), PCL, parallel CNN-LSTM residual (PCLR), and PCLA. In this study, we propose a combined model PCLRA to extract spatiotemporal features from the CHP engine sensor log data, which is a multivariate time series. CNN-, LSTM-, and CNN-LSTM-based sequential combination models are used as baselines to prove the performance. These models are benchmarks that have been applied to multivariate time series data prediction and anomaly detection in various fields and their performance is compared with the proposed model, PCLRA.

CNN is proposed to process multidimensional data such as images and videos. Since one-dimensional data are used as an input to the deep neural network, it is necessary to flatten 2D data, such as images, to one dimension, which results in a significant loss in the spatial features of the data. However, CNN learns spatial features without loss using 2D data as an input and they achieve excellent performance.

LSTM is proposed to address the vanishing gradient problem of RNN. RNN is structured to reflect the previous trained results in the current time and it is suitable for learning temporal features of the data. In the process of reflecting the previous trained results, LSTM uses three gates to select the parts to remember, delete, and add, and it reflects these parts in subsequent training.

Hybrid models that incorporate LSTM and CNN in various structures are also tested as baselines. In SCL, CNN and LSTM are arranged in series; the input data are entered into the CNN and the corresponding output is entered into the LSTM to obtain the final output. In SCLR, the loss of information is overcome by combining each CNN and LSTM of SCL with residual blocks. In SCLA, the attention mechanism is added to SCL to evaluate the importance of features in the LSTM output, and in SCLRA model, SCL is combined with both the residual blocks and attention mechanism.

In PCL, CNN and LSTM are arranged in parallel and two networks receive the same data as an input. Then, the final output is derived by combining the spatial and temporal features extracted from CNN and LSTM, respectively. In PCLR, the loss of information is supplemented by combining each CNN and LSTM of PCL with residual blocks, and the PCLA model derives the final output by applying the spatiotemporal features combined in PCL model to the attention mechanism.

### 2.8. Hyperparameter Optimization

In this study, optimal models are derived by applying BO to 10 models. BO consists of a surrogate model and an acquisition function. The surrogate model estimates an objective function using the Gaussian process based on the result of a previous experiment, and the acquisition function recommends the subsequent input value based on the estimation model. Through this, it is possible to obtain the combination of hyperparameters that optimizes the performance of the deep learning model.

## 3. Experiments

### 3.1. Data Description

The CHP engine sensor log data are collected from three CHP engines at a power plant that supply electricity and heating to approximately 12,000 households in Chungcheongnam-do, South Korea. Of the three engines, Engines 1 and 2 were introduced in 2009 and Engine 3 was introduced in 2015, and the engines are WÄRTSILÄ products. All three engines have operated continuously to the present day. The actual operation data per minute were collected from 07:00 on 3 May 2019 to 16:19 on 31 August 2020.

In this study, the engines are divided into five parts for anomaly detection according to the advice of power plant experts, as shown in Figure 5: the fuel gas part (FG), lube oil part (LO), charge air and exhaust gas part (CE), gas engine part (GE), and cooling water part (CW) [40,41]. Table 1 presents engine sensor feature categories for each part. As the part related to fuel and combustion, FG consists of combustors that burn LNG and includes sensor features related to the main duration offset, ignition timing, and knocking of the cylinder. LO is related to engine operation and is the part that operates the turbine and engine with the combusted LNG. It primarily includes sensor features related to the liner temperature, main bearing temperature, and temperature and pressure of the lube oil. CE is an exhaust gas-related part that discharges the burned LNG as waste gas via the heat exchanger. It includes sensor features related to the exhaust gas temperature of the cylinders, exhaust gas temperature deviation, boiler, and exhaust gas waste gate valve position following combustion. GE encompasses the engine, turbine, and generator and includes sensor features related to the engine speed and load, power and phase current of the phasor measurement unit, the district heater (DH), and the power, phase current, winding temperature, and bearing temperature of the generator. CW is the part that converts cold water into warm water via the combustor and heat exchanger. It includes sensor features related to the gas temperature and gas pressure of the LNG fuel and the dew point and temperature of the charge air cooler.

There is no separate category for anomaly type in the engine anomaly and management history data; instead, the abnormal engine, part, date and time of occurrence, and details of the action are recorded manually. The engine anomaly detection target anomalies that occur when the engine is in operation, that is, when the engine speed is 10 rpm or higher. Anomalies during engine shutdown can be excluded from detection because they are part of regular maintenance or inspection and repair and are used to address the anomalies during operation. Figure 6 shows 60 feature log datapoints of Engine 2 LO and the presence of anomalies during the month of September 2019 as an example. The figure indicates the occurrence of anomalies during engine operation on 16 September and during the engine shutdown on 7, 8, and 16 September.

The anomaly rates for each engine and part are presented in Table 2. Since the data have severe class imbalance, the macro f1 score is used as a model performance evaluation metric in this study.

Normalization is applied to convert the data to the range of 0–1 for model training. Min–max scaling [42] is then applied, as shown in Equation (15). We divided the difference between X and $X_{min}$ by the difference between $X_{max}$ and $X_{min}$, which yield $X^{\prime }$, i.e., the normalization result of X.
(15)$$X^{\prime }=\left(X-X_{min}\right)/\left(X_{max}-X_{min}\right)$$

### 3.2. Anomaly Detection Model Train and Test Procedures

The experimental environment consists of an Intel(R) Xeon(R) Silver 4210R CPU @ 2.40 GHz, 64 GB RAM, Windows 10 64 bit, and an NVIDIA GeForce RTX 3080 Ti. Python 3.8, TensorFlow 2.3, scikit-learn 0.23, and scikit-optimization 0.8 are used. The dataset is divided into training, validation, and test sets at a ratio of 6:2:2. Typically, the train and test set ratio of 8:2 is used, but in this study, model validation is performed during the training process, which is generally performed by splitting a portion of the train set. And, the three data sets are divided equally at normal anomaly ratios, but if they are not divided at the same ratio and are concentrated on one set, it is difficult to train, validate, and test the anomalies fairly.

The input time step of the anomaly detection model is 5 min, which is equally applied to all 10 models for 15 cases, and engine anomalies are detected before 2 min. This means that, for example, if the data from 20 October 2022 00:30:00 to 20 October 2022 00:34:00 is input to the model, an abnormal status of 20 October 2022 00:36:00 will be detected. At the power plant, the engine sensor log data is used in conjunction to manage the operation of the engines second-by-second. Therefore, the detection of anomalies 2 min after the data input can be useful for operating the engines and managing anomalies. The specifications for engine operation are based on guidance from the engine experts at the power plant. Including PCLRA, 10 deep learning models are trained for each engine and part.

BO is performed 50 times for each model and the criterion for searching is the macro f1 score, which is described in the next section. This means optimizing the hyperparameter in the direction of increasing this value, based on the macro f1 score. Via optimization, the final model is generated by selecting the model hyperparameters with the largest macro f1 score for the validation set and it is then trained and tested. BO is applied to determine the five hyperparameters, i.e., the number of output nodes and activation functions of the first and second layers, learning rate, and batch size that optimize the performance of the models, and Table 3 presents the search ranges.

### 3.3. Evaluation Criteria

The performances of the anomaly detection models are quantified and compared with regard to the macro f1 score, anomaly f1 score, and accuracy [43]. Table 4 presents the confusion matrix, which displays the anomaly detection results of the model. The number of abnormal cases that are classified accurately is referred to as true positive (TP), and true negative (TN) represents the number of normal cases accurately detected by the model. False positive (FP) represents the number of normal cases detected as an anomaly, and false negative (FN) represents the number of abnormal cases detected as normal. The accuracy refers to the ratio of the number of cases with accurate detection to the total number of cases, as given via Equation (16). However, in the case of imbalanced class data, the accuracy is biased toward the majority class, making it inappropriate as a performance evaluation criterion. Therefore, the model performance is evaluated via the f1 score using precision and recall. Precision is the ratio of the number of correctly detected cases to the number of abnormal results detected by the model, as given via Equation (17), and recall is the ratio of the number of abnormal cases detected by the model to the number of actual abnormal cases, as given via Equation (18). The f1 score, which is the harmonic mean of the precision and recall, is calculated using Equation (19) to identify the model with the best precision and recall values. This is referred to as the anomaly f1 score and is based on anomaly detection. The f1 score is calculated according to normal detection, and the macro f1 score, which is the unweighted mean of the normal f1 score and anomaly f1 score, can be calculated and used as a measure of the model performance.
(16)$$Accuracy=\frac{TP+TN}{TP+FP+TN+FN}$$
(17)$$Precision=\frac{TP}{TP+FP}$$
(18)$$Recall=\frac{TP}{TP+FN}$$
(19)$$F1-score=2\times \frac{precision\times recall}{precision+recall}$$

### 3.4. Anomaly Detection Performance Evaluation of Models

The 10 models are trained and tested for 15 cases, corresponding to three CHP engines and five parts. The results for the macro f1 score, anomaly f1 score, and accuracy are compared in Table 5, Table 6 and Table 7, respectively.

PCLRA, which is proposed as an anomaly detection model for CHP engines in this study, combines PCL with the residual blocks and attention mechanism, as shown in Figure 4. Two CNN layers and two LSTM layers are arranged in parallel and spatiotemporal features are extracted using the same input data. During this process, residual blocks are used to compensate for the loss of information caused by the vanishing gradient problem. The outputs from the CNN and LSTM are combined, the attention mechanism is used for training with a focus on the important features, and the occurrence of anomalies is derived as the output using the softmax function.

The statistics of the test results for 10 models based on the 5 five parts of three engines are compared, and the proposed PCLRA model exhibited the best performance. For the macro f1 score, it had the highest mean value (0.951) and the smallest standard deviation (std) (0.033), indicating its excellent performance. Additionally, for the anomaly f1 score, it had the highest mean value (0.903) and the smallest std (0.064). It also had the highest accuracy: 0.999 ± 0.002 (mean ± std). Among the 10 models, PCLRA performs the best for the LO, CE, and CW of Engine 1; FG, LO, and CW of Engine 2; and FG of Engine 3, and the second-best for the FG of Engine 1 and GE of Engine 2. An anomaly detection example of PCLRA targeting Engine 2 LO is shown in Figure 7. Abnormal symptoms occurred in Engine 2 LO from 13:33, and PCLRA detected that anomalies would occur at 13:31 (2 min prior).

Table 8 presents the hyperparameters obtained by optimizing the PCLRA model for the 15 cases. For example, in the case of the Engine 1 FG, the number of output nodes of the first CNN and LSTM layers is set to 21, that of the second layers is set to 13, and the activation function of each layer is set to tanh. The number of output nodes of the second layer is set to 13, which is identical to the number of output nodes of residual blocks. The learning rate of the model is 0.0069 and the batch size is 3533.

The confusion matrices of the best and second-best models for 15 cases are compared in Table A1. Each confusion matrix title indicates the engine, part, and model. The models that had the same performance exhibited the same confusion matrix; therefore, only one of the highest-performance models and one of the second-highest performance models are selected and compared. The proposed PCLRA model is one of the top two models in 9 of the 15 cases and it exhibited the best performance in seven cases. In the case where PCLRA is the second-best model, the normal misclassification rate of PCLRA is high for Engine 1 FG, whereas for Engine 2 GE, the normal misclassification rate of PCLRA is low, but the anomaly misclassification is high. As with Engine 1 LO, the white area in the confusion matrix heat map represents 0.

### 3.5. Performance for Different Methods of Combining CNN and LSTM

By training and testing 10 models for a total of 15 cases, the means and stds of the macro f1 score, anomaly f1 score, and accuracy are obtained. They are compared in Figure 5 and Figure 6. CNN–LSTM hybrid models can be divided into parallel models and serial models according to how CNN and LSTM are combined. In this study, to evaluate the two combination methods, PCL, SCL, and the modified structures of the two models are compared. Both PCL and SCL perform worse than uncombined CNN and LSTM models because the information loss due to the vanishing gradient problem increases in the hybrid model. However, PCL outperforms SCL by 8.93%, 20.45%, and 2.41% with regard to the macro f1 score, anomaly f1 score, and accuracy, respectively. Comparing SCLR and PCLR, which combine residual blocks with SCL and PCL, reveals that the macro f1 score and anomaly f1 score of PCLR are superior to those of SCLR by 3.83% and 8.74%, respectively; however, the accuracy of SCLR is 0.10% higher than that of PCLR. Comparing SCLA and PCLA, which included an attention mechanism, revealed that the macro f1 score of PCLA is 3.69% better and that the anomaly f1 score of PCLA is 8.56% better. Comparing SCLRA and PCLRA, which integrated both the residual block and attention mechanism into SCL and PCL, reveals that the macro f1 score, anomaly f1 score, and accuracy are 7.78%, 14.17%, and 3.30% higher, respectively, for PCLRA. The comparison results for the various serial and parallel combined models of CNN and LSTM confirm that parallel models more effectively extract and train the spatiotemporal features of multivariate time series data.

### 3.6. Performance for Different Information Loss Compensation Methods

In this study, the residual blocks and attention mechanism are combined with PCL and SCL to compensate for the information loss and their performance is compared in Figure 8 and Figure 9. Comparing SCLR and SCLA revealed that SCLR increased the macro f1 score by 2.22% and the anomaly f1 score by 5.21%. Comparing PCLR and PCLA revealed that PCLR increases the macro f1 score by 2.36% and the anomaly f1 score by 5.40%, but the accuracy of PCLA is 0.10% higher than that of PCLR. This indicates that residual blocks, which supplement the important information loss in the CNN and LSTM, contributes to the performance improvement of the model more than the attention mechanism. Finally, when both the residual blocks and attention mechanism are combined with the SCL, the macro f1 score of SCLRA increased by 2.62% and its anomaly f1 score increased by 8.39%, but its accuracy decreased by 3.11%. PCLRA performed the best with increases of 6.62%, 13.84%, and 0.20% in the macro f1 score, anomaly f1 score, and accuracy, respectively.

## 4. Discussion

In this section, issues related to the model proposed in this study are discussed.

First, this study uses the CNN-LSTM-based model structure to train spatiotemporal features of multivariate time series data and compares the performance by combining residual blocks and attention mechanisms to complement the performance. Experimental results show that CNN and LSTM have superior performance compared to simply combined models like SCL and PCL, but single models have the limitation of not being able to train spatiotemporal features. Therefore, this study aims to improve the structure of CNN-LSTM-based for training spatiotemporal features of multivariate time series data and proves the superior performance of PCLRA. Structures that combine the residual blocks and attention mechanism in a single model of CNN and LSTM have limitations in training spatiotemporal features, but experiments on these are considered necessary and will be researched.

In addition, this study used BO to derive the best performance of 10 models and compare them. There is a need to study how various hyperparameters affect performance in the HO process and improve search more efficiently based on this.

We will continue to resolve these issues that need to be considered and find ways to improve it.

## 5. Conclusions

The proposed PCLRA for CHP engine anomaly detection is a hybrid deep learning model that combines the CNN, LSTM, residual blocks, and attention mechanism. The engine sensor log data, in the form of a multivariate time series, are entered into the CNN and LSTM in parallel to extract spatial and temporal features. During this process, residual blocks are used to compensate for the important features that are lost in the CNN and LSTM for model training. Finally, the spatiotemporal features extracted from CNN and LSTM are input into the attention mechanism to focus on important features, and the probability of an engine anomaly is derived as an output using the softmax function. The performance of the proposed model is demonstrated by comparing the model to nine other models: CNN, LSTM, SCL, SCLR, SCLA, SCLRA, PCL, PCLR, and PCLA. The optimal performance of each model is determined and the performances are compared by applying BO to the 10 models.

Three CHP engines are used for the experiment and anomaly detection models are trained and tested on the five parts of each engine: FG, LO, CE, GE, and CW. The performance of the 10 models is compared for all 15 cases. First, we compare SCL, PCL, and modify models that combine CNN and LSTM in different ways. The serially combined models SCL, SCLR, SCLA, and SCLRA exhibited inferior anomaly detection performance to the parallelly combined models PCL, PCLR, PCLA, and PCLRA. The results confirm that serially combined models derive and train the spatiotemporal features of multivariate time series data much better in parallelly combined models. Next, we compare models that integrate the residual blocks and attention mechanism to improve the performance of SCL and PCL, which perform worse than the uncombined CNN and LSTM. SCLR and PCLR, which integrate residual blocks, exhibit superior anomaly detection performance to SCLA and PCLA, which integrate the attention mechanism. This result confirms that the method of training important features by combining residual blocks with CNN and LSTM instead of using the attention mechanism to train important spatiotemporal features extracted from the CNN and LSTM improves the anomaly detection performance. Furthermore, better performance is obtained when both the residual blocks and attention mechanism are used. Accordingly, PCLRA, which combines CNN and LSTM in parallel and uses both the residual blocks and attention mechanism, achieves the best performance among the 10 models, with a macro f1 score of 0.951 ± 0.033, an anomaly f1 score of 0.903 ± 0.064, and an accuracy of 0.999 ± 0.002. The model does not exhibit the best performance for all 15 cases; however, its performance is consistently superior to that of the other models, regardless of the engine and part. The statistics calculated for the model performance in the 15 cases based on the engines suggest that PCLRA performs the best for the old Engines 1 and 2, which have a long operating time, and the second best for Engine 3, which has a shorter operating time. The statistics calculated for the model performance in the 15 cases based on the parts indicate that PCLRA performs the best for the LNG combustion related FG, LO, and CE and the second best for CW.

The proposed PCLRA model for CHP engine anomaly detection achieves excellent performance, and we expect that it can be utilized at power plants to enhance engine stability and efficiency. In the future, we plan to expand our research on CHP anomaly detection by collecting data over a long-term period and further subdividing the types of anomalies experienced by CHP engines.

## Figures and Tables

**Figure 1 sensors-23-08746-f001:**
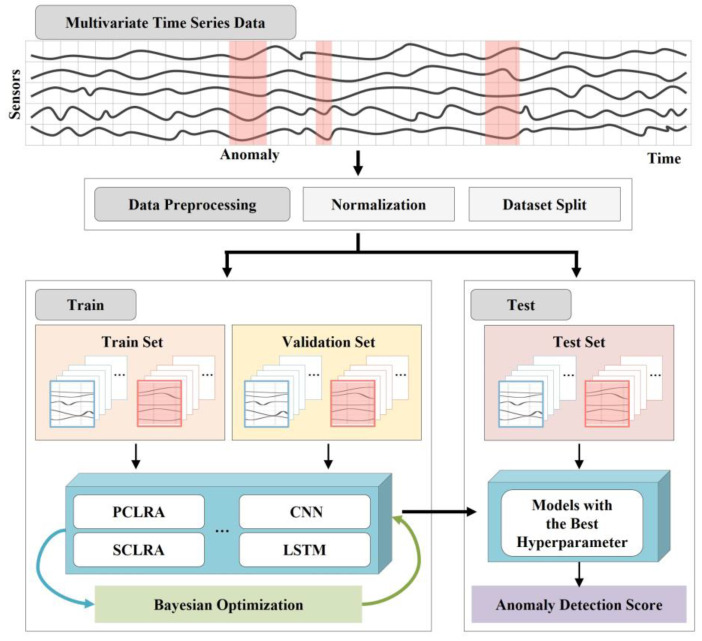
Framework of Multivariate Time Series Anomaly Detection.

**Figure 2 sensors-23-08746-f002:**
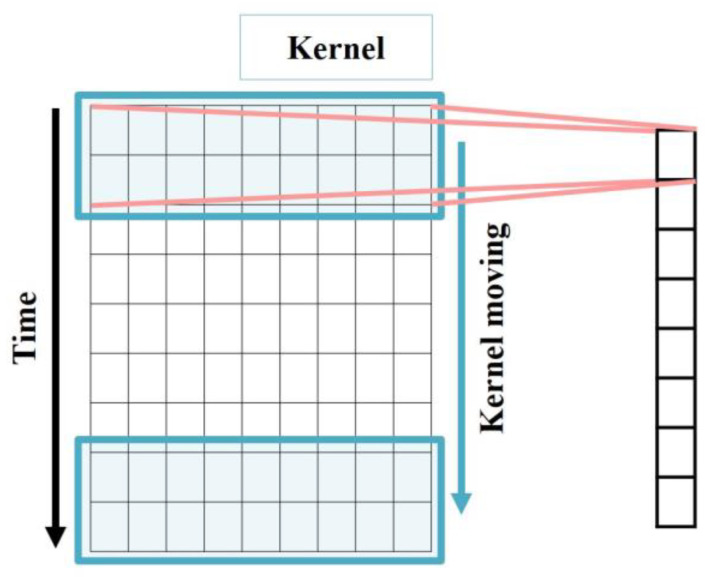
Structure of 1D-CNN.

**Figure 3 sensors-23-08746-f003:**
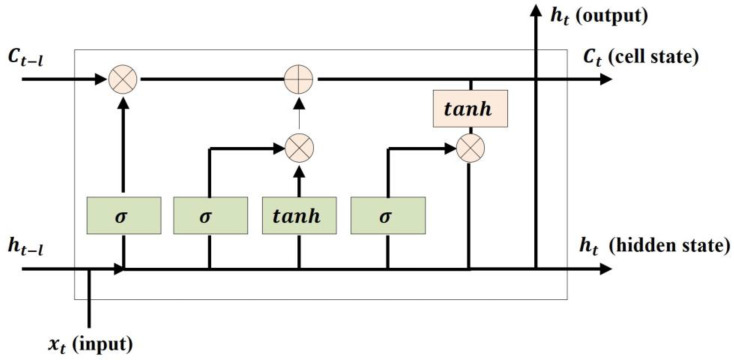
Structure of LSTM Cell.

**Figure 4 sensors-23-08746-f004:**
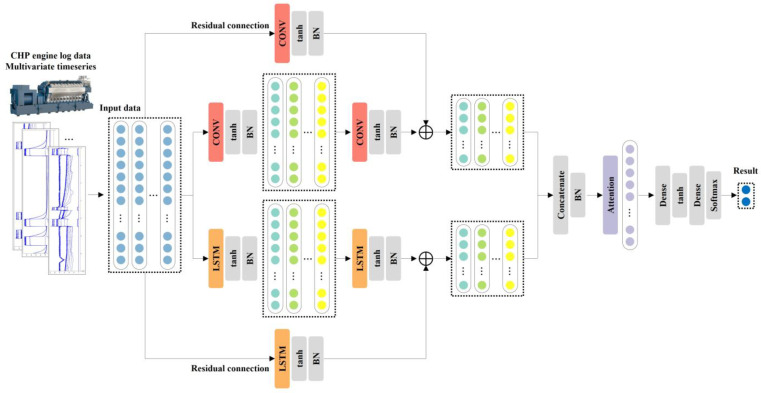
PCLRA model architecture.

**Figure 5 sensors-23-08746-f005:**
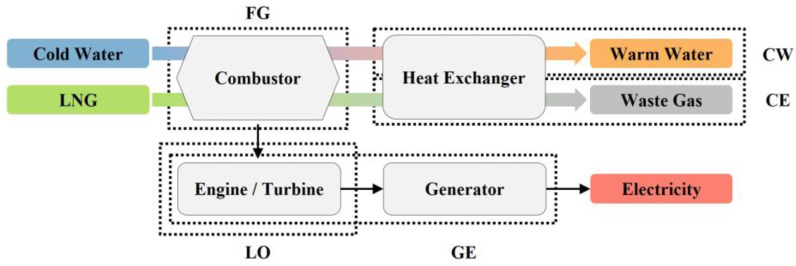
Five parts of the CHP engines.

**Figure 6 sensors-23-08746-f006:**
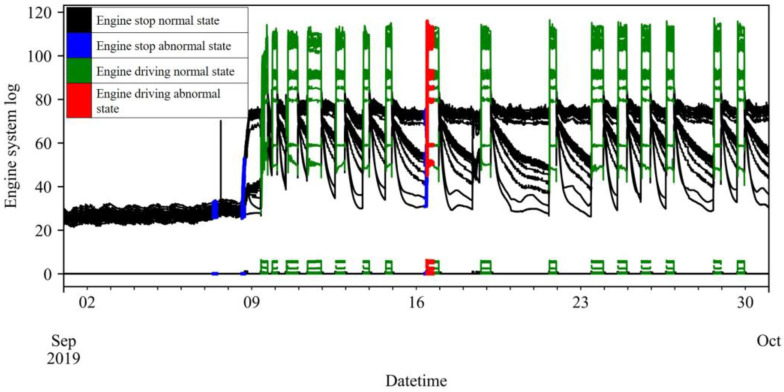
Engine 2 LO part log data example.

**Figure 7 sensors-23-08746-f007:**
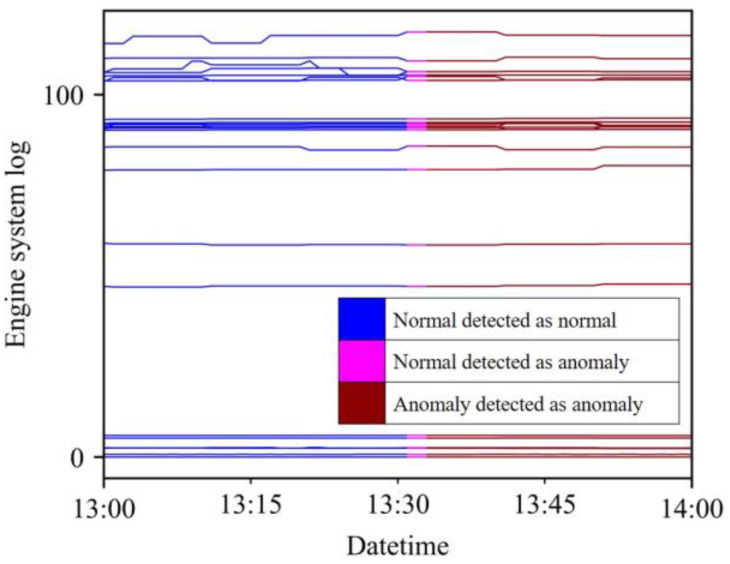
PCLRA anomaly detection result example for the Engine 2 LO.

**Figure 8 sensors-23-08746-f008:**
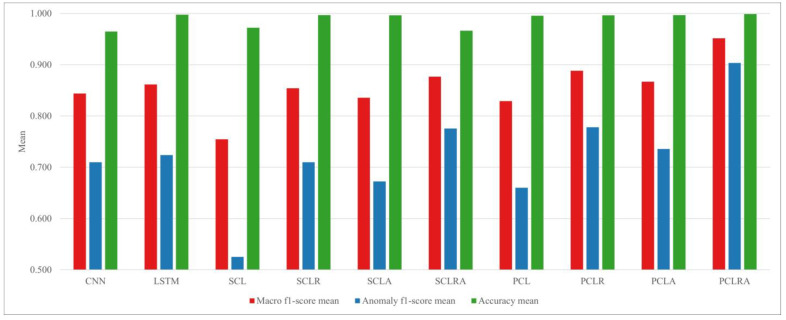
Performance comparison of anomaly detection models.

**Figure 9 sensors-23-08746-f009:**
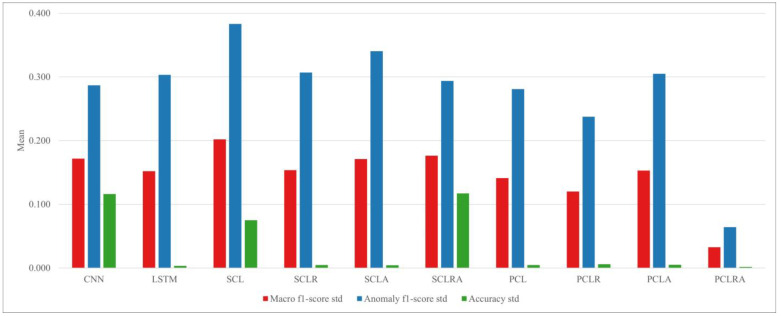
Standard deviation comparison of anomaly detection models.

**Table 1 sensors-23-08746-t001:** Sensor features by engine parts.

Part	Feature Category	Count	Total	Part	Feature Category	Count	Total
FG	Main duration offset	21	64	CE	Exhaust gas temperature	23	51
	Ignition timing	20			Exhaust gas temp deviation	20	
	Knocking	20			Boiler/DH	4	
	Others	2			Exhaust gas waste gate	3	
	Fuel	1			Others	2	
LO	Liners	40	60	GE	Load and speed	18	39
	Bearings	12			Boiler/DH	14	
	Lube oil	6			Generator	5	
	Others	2			Others	2	
	Boiler/DH	1		CW	Fuel	9	17
					Charge air	5	
					Others	2	
					Cooling	1	

**Table 2 sensors-23-08746-t002:** Anomaly rates (%) of 15 data sets corresponding to three engines and five parts.

	FG	LO	CE	GE	CW
Engine 1	0.97	0.62	0.15	0.78	0.15
Engine 2	0.63	0.36	0.31	0.31	0.31
Engine 3	0.83	0.34	0.28	0.28	0.28

**Table 3 sensors-23-08746-t003:** BO search range of 10 models.

Hyperparameter	Range
Learning rate	[1 × 10^−4^, 1 × 10^−2^]
Output nodes of first layer	[16, 32]
Output nodes of second layer	[8, 16]
Activation function	{‘tanh’, ‘relu’}
Batch size	[500, 5000]

**Table 4 sensors-23-08746-t004:** Confusion matrix.

		Predicted	
		Anomaly	Normal
Actual	Anomaly	TP	FN
	Normal	FP	TN

**Table 5 sensors-23-08746-t005:** Anomaly detection model macro f1 score.

	Engine	Part	CNN	LSTM	SCL	SCLR	SCLA	SCLRA	PCL	PCLR	PCLA	PCLRA
Macro F1 score	1	FG	**0.883**	0.834	0.723	0.752	0.834	0.844	0.752	0.835	0.742	0.858
		LO	0.348	0.500	0.500	0.500	0.500	0.347	0.573	0.522	0.500	**0.962**
		CE	**0.964**	0.882	0.782	0.937	**0.964**	0.564	0.657	0.855	0.948	**0.964**
		GE	0.800	0.498	0.498	0.926	0.527	**0.990**	**0.990**	0.987	0.536	0.978
		CW	0.552	**0.964**	0.500	0.500	0.523	0.937	0.601	0.818	0.964	**0.964**
	2	FG	0.847	0.811	0.806	0.794	0.834	0.843	0.814	0.765	0.841	**0.946**
		LO	0.982	0.974	0.422	0.977	**0.984**	0.977	0.982	0.982	0.982	**0.984**
		CE	0.968	0.970	0.928	0.960	**0.985**	0.982	0.728	0.976	0.913	0.976
		GE	0.915	0.970	0.522	0.973	**0.976**	**0.976**	0.973	**0.976**	**0.976**	**0.976**
		CW	0.976	0.973	0.976	0.893	**0.979**	**0.979**	**0.979**	0.973	**0.979**	**0.979**
	3	FG	0.820	0.821	0.849	0.816	0.875	0.872	0.842	0.841	0.809	**0.908**
		LO	0.957	0.924	0.959	0.950	0.953	0.956	0.935	0.954	**0.971**	0.919
		CE	0.961	0.938	**0.965**	0.948	0.842	0.951	0.735	0.938	0.961	0.958
		GE	0.908	0.938	0.954	0.948	0.954	**0.969**	0.929	0.952	0.936	0.951
		CW	0.774	0.922	0.936	0.938	0.800	**0.965**	0.942	0.948	0.948	0.947
mean			0.844	0.861	0.755	0.854	0.835	0.877	0.829	0.888	0.867	**0.951**
std			0.172	0.152	0.202	0.154	0.171	0.176	0.141	0.120	0.153	**0.033**

Bold score means excellent performance in each engine and part.

**Table 6 sensors-23-08746-t006:** Anomaly detection model anomaly macro f1 score.

	Engine	Part	CNN	LSTM	SCL	SCLR	SCLA	SCLRA	PCL	PCLR	PCLA	PCLRA
Anomaly F1 score	1	FG	**0.769**	0.672	0.455	0.513	0.672	0.692	0.513	0.674	0.493	0.720
		LO	0.003	0.000	0.000	0.000	0.000	0.003	0.150	0.055	0.000	**0.923**
		CE	**0.929**	0.765	0.565	0.875	**0.929**	0.133	0.317	0.710	0.897	**0.929**
		GE	0.605	0.000	0.000	0.854	0.061	**0.981**	**0.981**	0.974	0.075	0.955
		CW	0.115	**0.929**	0.000	0.000	0.049	0.875	0.205	0.636	**0.929**	**0.929**
	2	FG	0.697	0.626	0.615	0.592	0.671	0.688	0.631	0.536	0.684	**0.893**
		LO	0.963	0.948	0.023	0.953	**0.969**	0.953	0.964	0.964	0.964	**0.969**
		CE	0.936	0.940	0.857	0.920	**0.970**	0.964	0.460	0.953	0.827	0.952
		GE	0.831	0.940	0.080	0.947	**0.953**	**0.953**	0.947	**0.953**	**0.953**	0.952
		CW	0.953	0.947	0.953	0.786	**0.959**	**0.959**	**0.959**	0.947	**0.959**	**0.959**
	3	FG	0.644	0.646	0.702	0.637	0.754	0.746	0.688	0.686	0.623	**0.818**
		LO	0.914	0.849	0.919	0.901	0.907	0.913	0.871	0.908	**0.943**	0.839
		CE	0.923	0.876	0.930	0.896	0.686	0.902	0.472	0.876	0.923	0.916
		GE	0.816	0.876	0.909	0.896	0.909	**0.938**	0.859	0.904	0.871	0.902
		CW	0.550	0.845	0.871	0.876	0.601	**0.930**	0.884	0.897	0.896	0.894
mean			0.710	0.724	0.525	0.710	0.673	0.775	0.660	0.778	0.736	**0.903**
std			0.287	0.303	0.383	0.307	0.341	0.294	0.281	0.237	0.305	**0.064**

Bold score means excellent performance in each engine and part.

**Table 7 sensors-23-08746-t007:** Anomaly detection model accuracy.

	Engine	Part	CNN	LSTM	SCL	SCLR	SCLA	SCLRA	PCL	PCLR	PCLA	PCLRA
Accuracy	1	FG	**0.994**	0.991	0.981	0.982	0.991	0.992	0.982	0.991	0.981	0.993
		LO	0.530	0.999	0.999	0.999	0.999	0.529	0.993	0.979	0.999	**1.000**
		CE	**1.000**	0.999	0.998	**1.000**	**1.000**	0.991	0.994	0.999	**1.000**	**1.000**
		GE	0.990	0.992	0.992	0.997	0.984	**1.000**	**1.000**	1.000	0.992	0.999
		CW	0.979	**1.000**	0.999	0.999	0.992	**1.000**	0.994	0.999	1.000	**1.000**
	2	FG	0.995	0.993	0.992	0.992	0.994	0.995	0.993	0.990	0.995	**0.999**
		LO	**1.000**	**1.000**	0.699	**1.000**	**1.000**	**1.000**	**1.000**	**1.000**	**1.000**	**1.000**
		CE	**1.000**	**1.000**	0.999	0.999	**1.000**	**1.000**	0.993	**1.000**	0.999	**1.000**
		GE	0.999	**1.000**	0.932	**1.000**	**1.000**	**1.000**	**1.000**	**1.000**	**1.000**	**1.000**
		CW	**1.000**	**1.000**	**1.000**	0.998	**1.000**	**1.000**	**1.000**	**1.000**	**1.000**	**1.000**
	3	FG	0.991	0.991	0.993	0.991	0.995	0.995	0.993	0.993	0.990	**0.997**
		LO	0.999	0.999	0.999	0.999	0.999	0.999	0.999	0.999	1.000	0.999
		CE	**1.000**	0.999	**1.000**	0.999	0.998	0.999	0.994	0.999	1.000	**1.000**
		GE	0.999	0.999	0.999	0.999	0.999	**1.000**	0.999	0.999	0.999	0.999
		CW	0.996	0.999	0.999	0.999	0.997	**1.000**	0.999	0.999	0. 999	0.999
mean			0.965	0.997	0.972	0.997	0.997	0.963	0.996	0.996	0.997	0.999
std			0.116	0.003	0.075	0.005	0.004	0.117	0.005	0.006	0.005	0.002

Bold score means excellent performance in each engine and part.

**Table 8 sensors-23-08746-t008:** PCLRA anomaly detection model hyperparameter results.

Engine	Part	Learning Rate	Output Nodes 1	Output Nodes 2	Activation Function	Batch Size
1	FG	0.0069	21	13	tanh	3533
	LO	0.0015	25	11	relu	3498
	CE	0.0013	17	11	tanh	3559
	GE	0.0003	29	12	relu	5000
	CW	0.0033	31	8	tanh	1222
2	FG	0.0030	16	15	relu	500
	LO	0.0023	16	10	relu	562
	CE	0.0006	25	9	tanh	858
	GE	0.0008	24	10	relu	857
	CW	0.0009	31	8	relu	4156
3	FG	0.0046	30	8	tanh	4973
	LO	0.0079	16	16	relu	5000
	CE	0.0005	21	12	tanh	1972
	GE	0.0091	32	12	tanh	4916
	CW	0.0095	30	9	tanh	5000

## Data Availability

The authors do not have permission to disclose the data.

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
