# Peer review of "CHP Engine Anomaly Detection Based on Parallel CNN-LSTM with Residual Blocks and Attention"

_sensors, 2023, doi:10.3390/s23218746_

Round 1
Reviewer 1 Report
Comments and Suggestions for Authors
Dear authors,
I have reviewed your manuscript titled "Multivariate Time Series Anomaly Detection in CHP Engine Systems" as a scientific reviewer for the Sensors Journal. I commend you for your valuable contributions to the field of anomaly detection in engine systems. However, the following comments need to be addressed before considering your manuscript for publication in the journal.
- While the authors provide excellent context for the research, it might be helpful to provide brief definitions or explanations of terms like "CHP engine" and "anomaly detection."
- Although Bayesian optimization for hyperparameter tuning is mentioned, it would be advantageous to include additional elaboration regarding the particular hyperparameters tuned and the range of values examined.
- The introduction of baseline models is informative and helps understand the comparative analysis. It is important to clarify that these models serve as benchmarks for evaluating the performance of your proposed model.
- The authors used Bayesian optimization for hyperparameter tuning, which is a good approach, and its explanation is clear. However, the authors might consider providing context on the range or limits of hyperparameter values explored during optimization.
- The authors could briefly discuss why they chose specific components and models for their methodology. Explain why CNN and LSTM were chosen over other architectures and why the authors integrated residual blocks and an attention mechanism.
- The authors should Consider discussing the sensitivity of their model to hyperparameter choices. They could mention if any sensitive hyperparameters significantly influenced performance.
- The authors should provide details on their validation strategy, including the criteria used to select the best hyperparameter combinations during training and validation.
- If applicable, the authors could indicate whether the source code for their methodology is accessible to the public or can be shared with scholars who express interest. Implementing this approach can enhance the reproducibility of findings and facilitate future investigations.
- The authors mentioned that the dataset is divided into training, validation, and test sets in a 6:2:2 ratio with identical anomaly rates. The authors should briefly explain why this specific ratio and balancing strategy were chosen.
- The authors mentioned that engine anomalies are detected before 2 minutes. It would be helpful to clarify what this means in the context of the author's study. Does it refer to the time window before an anomaly occurs?
- The authors should explain why precisely 10 deep-learning models were trained for each engine and part. Was this number chosen based on empirical observations or any specific rationale?
Comments on the Quality of English Language
The quality of the English language in the manuscript is Moderate editing of the English language required. Some sentences and phrases could be improved for clarity and readability. Overall, the language is understandable, but it would benefit from some additional editing for smoother flow and precision.
Author Response
I appreciate the time and effort that you have dedicated to providing your valuable feedback on my manuscript.
Please see the attachment.

Reviewer 2 Report
Comments and Suggestions for Authors
In this paper, the authors propose a parallel convolutional neural network-long-term memory (CNN-LSTM) residual blocks attention (PCLRA) model designed for anomaly detection using CHP engine sensor data. The key idea is to use CNN and LSTM models concurrently to extract spatiotemporal features. This model is further enhanced with the inclusion of residual blocks and attention mechanisms. The proposed approach looks technically sound and the manuscript is well written. I agree that the proposed PCLRA achieved superior performance in all cases.
However, there is an issue in performance comparisons. When looking at Tables 5-7, it seems that the CNN and LSTM algorithms are not complementary. With a few exceptions, the performance difference between the two algorithms is not significantly large. This leads to an ambiguity regarding the rationale behind adopting a parallel configuration of CNN and LSTM to improve the performance. Therefore, I suggest that the authors undertake evaluations using variants of the CNN or LSTM algorithm, such as CR, LR, CA, RA, CRA or LRA, to provide a more comprehensive performance.
Author Response

(The authors gave the same response as above.)

Round 2
Reviewer 2 Report
Comments and Suggestions for Authors
The authors have adequately addressed the concerns raised by reviewers. Therefore, I recommend that this manuscript be accepted for publication in this journal.